# VIDEOEVAL: COMPREHENSIVE BENCHMARK SUITE FOR LOW-COST EVALUATION OF VIDEO FOUNDATION MODEL

## ABSTRACT

With the accumulation of high-quality data and advancements in visual pretraining paradigms, recent Video Foundation Models (VFMs) have made significant progress, demonstrating remarkable performance on popular video understanding benchmarks. However, conventional benchmarks (e.g. Kinetics) and evaluation protocols are limited by their relatively poor diversity, high evaluation costs, and saturated performance metrics. In this work, we introduce a comprehensive benchmark suite to address these issues, namely **VideoEval**. We establish the **Vid**eo **T**ask **A**daption **B**enchmark (**VidTAB**) and the **Vid**eo **E**mbedding **B**enchmark (**VidEB**) from two perspectives: evaluating the task adaptability of VFMs under few-shot conditions and assessing their feature embedding's direct applicability to downstream tasks. With VideoEval, we conduct a large-scale study of 20 popular open-source vision foundation models. Our study reveals some insightful findings, 1) overall, current VFMs exhibit weak generalization across diverse tasks, 2) increasing video data, whether labeled or in video-text pairs, does not necessarily improve task performance, 3) the effectiveness of some pre-training paradigms may not be fully validated in previous benchmarks, and 4) combining different pre-training paradigms can help develop models with better generalization capabilities. We believe this study serves as a important complement to the current evaluation methods for VFMs and offers valuable insights for future research directions.

## 1 INTRODUCTION

The field of deep learning is experiencing a significant paradigm shift due to the emergence of foundation models (FMs). These models, exemplified by BERT Devlin et al. (2018), GPT Brown et al. (2020); OpenAI (2023a;b), CLIP Radford et al. (2021) and Stable Diffusion Rombach et al. (2021), are trained on massive and diverse data at scale and demonstrate remarkable adaptability to a broad spectrum of downstream tasks.

In the realm of video understanding, early researchers train backbone networks Feichtenhofer et al. (2019); Bertasius et al. (2021); Liu et al. (2022); Fan et al. (2021) using visual classification tasks on large-scale labeled datasets like ImageNet Deng et al. (2009) and Kinetics Kay et al. (2017b). However, the high cost associated with labeled data promotes the development of self-supervised learning methods that capitalize on unlabeled data for visual pre-training Pan et al. (2021); Wei et al. (2022); Feichtenhofer et al. (2022); Tong et al. (2022); Wang et al. (2022a). Furthermore, researchers delve into multimodal pre-training utilizing large-scale visual-text pairs Xu et al. (2021a); Yan et al. (2022); Wang et al. (2024a); Li et al. (2023), thereby enhancing their models' capabilities and demonstrating impressive zero-shot performance. Overall, fueled by the accumulation of high-quality image and video data and advancements in visual pre-training paradigms, Video Foundation Models (VFMs) witness remarkable progress in recent years. A new generation of VFMs Feichtenhofer et al. (2022); Tong et al. (2022); Wang et al. (2023b); Bardes et al. (2023); Zhao et al. (2024); Wang et al. (2022b; 2024b) emerges, demonstrating outstanding performance on conventional video understanding benchmarks.

The rapid development of VFMs raises the problem: ***How to evaluate a video foundation model?*** In image realm, Previous works assess the generalization capability of Image Foundation Models

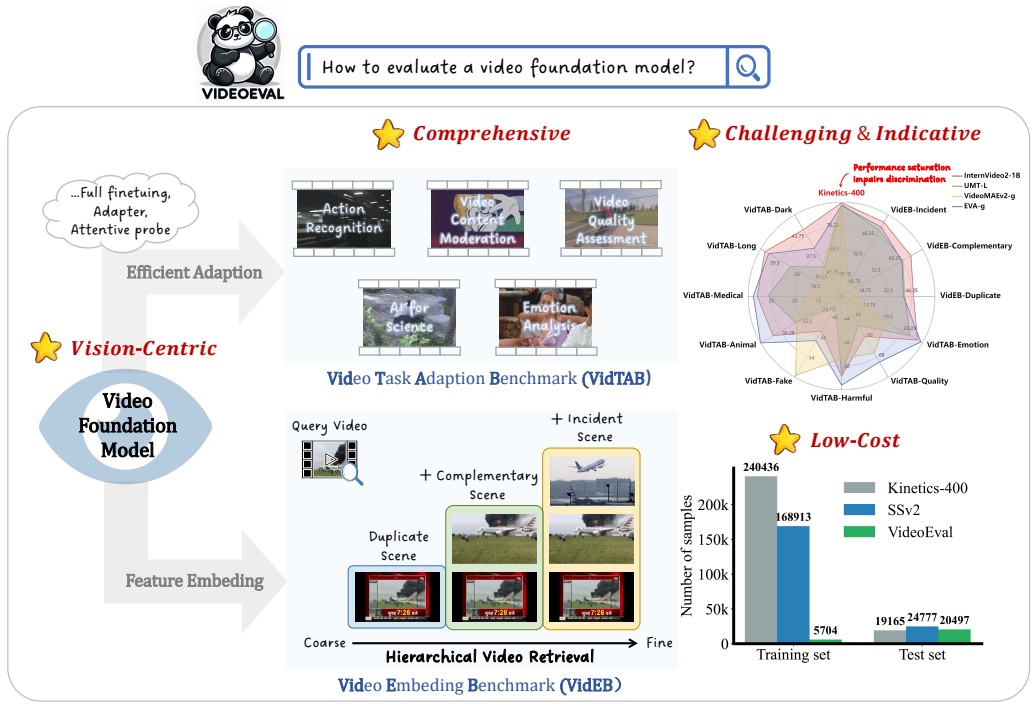

Figure 1: **Overview of VideoEval.** We propose a novel, vision-centric evaluation method for video foundation models that is comprehensive, challenging, indicative, and low-cost.

(IFMs) by evaluating their performance on numerous downstream visual tasks, encompassing diverse scenarios and evaluation protocols Zhai et al. (2019); Hendrycks et al. (2021b); Recht et al. (2019); Hendrycks et al. (2021a); Wang et al. (2019); Idrissi et al. (2023); Goldblum et al. (2023). However, previous works primarily evaluates VFMs through benchmarks focusing on action recognition tasks Tong et al. (2022); Bardes et al. (2023); Yuan et al. (2023). Some studies Wang et al. (2022b; 2024b); Zhao et al. (2024) have also considered combining language models to evaluate performance on multimodal tasks. There are **several problems with current evaluation methods**: **(1)** Benchmarks like Kinetics Kay et al. (2017b), Something Goyal et al. (2017b) and AVA Gu et al. (2017), which focus on action recognition, overlook other video understanding scenarios (e.g., video quality assessment), limiting their applicability in evaluating the generalization capabilities of visual foundation backbones across diverse video understanding applications. **(2)** The performance of VFMs on conventional benchmark Kay et al. (2017a) has reached a saturation point (90% Top-1 accuracy), making it challenging to differentiate between the true capabilities of different VFMs. **(3)** The high validation costs associated with conventional evaluation protocols, which often necessitate end-to-end training on the entire dataset, pose a significant challenge, particularly for large VFMs. **(4)** Incorporating language models may introduce bias when evaluating VFMs, as performance differences might stem from the language model rather than the VFMs itself.

To tackle these problems, we build a comprehensive benchmark suite for evaluation of VFMs, namely VideoEval. As shown in Figure 1, our method has the following key features: **_Comprehensive_**: First, we created the Video Task Adaptation Benchmark (VidTAB) to evaluate the adaptability of VFMs to unseen tasks with limited samples. We collected public datasets from various video task domains, including action recognition in special scenarios, AI for science, video content moderation, video quality/aesthetic assess, and emotion analysis. From these domains, we constructed eight adaptation tasks and developed evaluation protocols and adaptation methods suitable for current VFMs. Additionally, to assess the capability of VFMs' feature embedding for downstream applications, we created the Video Embedding Benchmark (VidEB), which includes four tasks that evaluate embedding at different granularities. **_Challenging & Indicative_**: Due to the diversity of test data and the effectiveness of our evaluation protocols, our VideoEval can effectively distinguish between various VFMs that perform similarly on traditional benchmarks, providing deeper insights into their true capabilities. **_Low-cost_**: Thanks to our training-light few-shot evaluation and training-free feature

embedding evaluation protocols, VideoEval requires significantly fewer training samples compared to previous benchmarks, while maintaining a comparable number of testing samples to ensure accurate and stable evaluations. ***Vision-centric***: Our evaluation focuses solely on the Video FMs themselves, avoiding the introduction of biases that may arise from incorporating language models.

Based on VideoEval, we evaluate 20 open-source vision foundation models, including VFMs, Image Foundation Models (IFMs), and IFMs with image-to-video methods. **Our main findings as following:** First, current VFMs still struggle to adapt to unseen video tasks with limited training samples. Second, while more data and larger models generally improve performance, augmenting video training data can sometimes negatively affect certain tasks. Third, the effectiveness of certain pre-training paradigms, such as VideoMAEv2 Wang et al. (2023b), may not have been adequately validated in previous benchmarks. Finally, combining multiple pre-training paradigms can lead to models with better generalization capabilities, such as performing multimodal contrastive learning after unimodal visual self-supervised pre-training Li et al. (2023); Wang et al. (2024b).

Table 1: **Comparison of VFMs Benchmark.** "Num. training" denotes number of training samples, "Num. test" denotes number of test samples, and "Beyond Action" denotes the tasks in this benchmark extend beyond action understanding. Compared to previous benchmarks, our VideoEval framework achieves more comprehensive and reliable evaluations at a lower cost.

| Benchmark | Num. training | Num. test | Beyond Action | Task Diversity | Domain Diversity | VFMs-specific protocol |
|---|---|---|---|---|---|---|
| *Single-dataset Benchmarks* | | | | | | |
| Kinetics-400 Kay et al. (2017a) | 240,436 | 19,165 | ✗ | ✗ | ✗ | ✗ |
| Sth-Sth V2 Goyal et al. (2017a) | 168,913 | 24,777 | ✗ | ✗ | ✗ | ✗ |
| Moment-in-Time Monfort et al. (2020) | 791,246 | 33,898 | ✗ | ✗ | ✗ | ✗ |
| UCF101 Soomro et al. (2012) | 9,537 | 3,783 | ✗ | ✗ | ✗ | ✗ |
| *Multi-dataset Benchmarks* | | | | | | |
| SEVERE Thoker et al. (2022b) | 868,446 | 144,830 | ✗ | ✓ | ✓ | ✗ |
| BEAR Deng et al. (2023) | 240,236 | 140,436 | ✗ | ✓ | ✓ | ✗ |
| VideoGLUE Yuan et al. (2023) | 1,896,621 | 239,011 | ✗ | ✓ | ✓ | ✓ |
| **VideoEval** | 5,704 | 20,497 | ✓ | ✓ | ✓ | ✓ |

## 2 RELATED WORK

**Video foundation models**  With the continuous growth of image Sharma et al. (2018); Changpinyo et al. (2021); Schuhmann et al. (2022) and video data Bain et al. (2021); Wang et al. (2024a); Chen et al. (2023; 2024a;b) and advancements in pre-training paradigms, research on Video Foundation Models (VFMs) has progressed rapidly. Current VFMs are primarily built around two pre-training paradigms: masked video modeling based on unimodal video data Feichtenhofer et al. (2022); Tong et al. (2022); Wang et al. (2023b; 2022a; 2023c); Girdhar et al. (2023); Ryali et al. (2023) and video-text contrastive learning based on multimodal visual-text pairs Xu et al. (2021a); Wang et al. (2023a); Yan et al. (2022); Cheng et al. (2022); Wang et al. (2024a). Some works Wang et al. (2022b); Li et al. (2023); Zhao et al. (2024) combine these paradigms, enabling VFMs to extend further into multimodal understanding. Additionally, some studies introduce modalities like audio and speech on top of video and text Chen et al. (2023; 2024a); Wang et al. (2024b), further expanding the capabilities of VFMs. Recently, InternVideo2 Wang et al. (2024b) leverages mature pre-training paradigms and large-scale high-quality data to scale VFMs to 6 billion parameters, achieving remarkable performance improvements.

**Evaluation of VFMs**  Previous works primarily utilize action recognition benchmarks focused on appearance and motion Kay et al. (2017b); Goyal et al. (2017a); Gu et al. (2017) to evaluate VFMs. To enhance evaluation diversity, some studies explore richer domains and tasks Thoker et al. (2022a); Deng et al. (2023); Schiappa et al. (2023), but they remain limited to action recognition tasks. The InternVideo series Wang et al. (2022b; 2024b) and VideoGLUE Yuan et al. (2023) attempt to provide a more comprehensive evaluation of VFMs by expanding the number of benchmarks and evaluation protocols. However, these efforts are still based on existing benchmarks and incurred high validation costs. In contrast, our work considers the characteristics and application scenarios of VFMs, offering a comprehensive and low-cost evaluation solution through task definition and evaluation protocols, aimed at rapidly verifying the generalization capabilities of VFMs—a crucial aspect currently lacking in the community's development of these models.

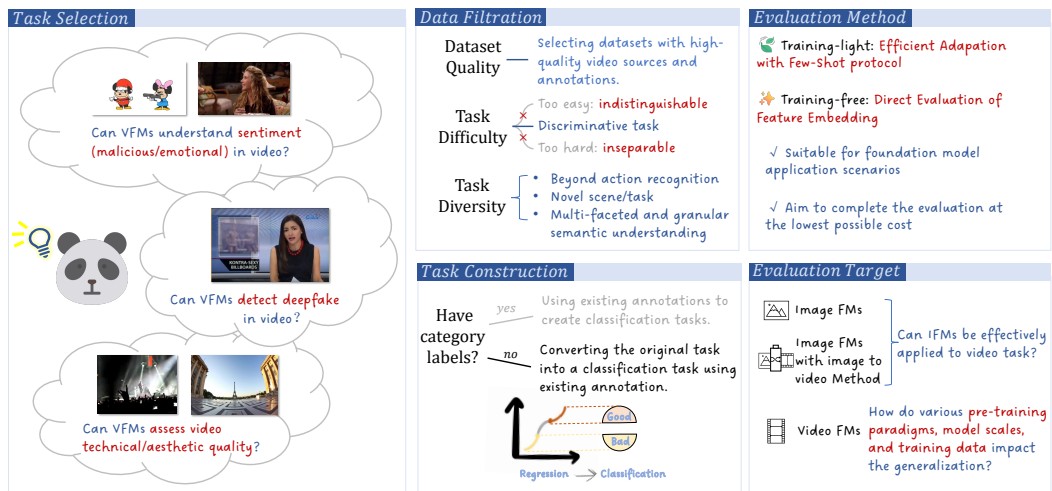

Figure 2: **Illustration of building VideoEval.**

## 3 BUILDING VIDEOEVAL

We argue that a powerful video foundation model should possess two key capabilities: (1) strong task adaptation ability, i.e., the ability to *adapt to diverse, unseen tasks with limited training samples*, and (2) the capacity to *extract feature embedding that retain and distill key information from videos*, directly supporting various downstream tasks. From these perspectives, we construct VideoEval, which includes the Video Task Adaptation Benchmark (VidTAB) and the Video Embedding Benchmark (VidEB). By creating diverse task scenarios and employing efficient evaluation methods, VideoEval can quickly and comprehensively assess the generalization ability of VFMs in video understanding. In this section, we present our VideoEval in detail. The construction pipeline for VideoEval is illustrated in Figure 2, and the evaluation tasks we ultimately constructed are presented in Table 2.

### 3.1 VIDEO TASK ADAPTION BENCHMARK

**Collecting diverse dataset from public source.** Previous benchmarks primarily focus on evaluating video models based on human actions, overlooking many other tasks requiring video understanding. Therefore, we consider five different application scenarios:

- *Action Recognition in Special Scenarios* (**Action**): While previous benchmarks have extensively examined action recognition tasks, our focus here is to assess VFMs' capabilities in recognizing actions within special scenarios.

- *AI for Science* (**Science**): Referencing previous work Zhao et al. (2024), we classify tasks related to medicine and natural sciences as a category.

- *Video Content Moderation* (**Safety**): We group tasks related to identifying harmful or misleading information in video content.

- *Video Quality Assessment* (**Quality**): We categorize more subjective tasks into this group. The goal is to assess VFMs' ability to learn low-level information and human aesthetic preferences.

- *Emotion Analysis* (**Emotion**): We group tasks related to human emotion analysis into this category to evaluate VFMs' ability to understand and analyze human emotions.

**Constructing the adaptation task based on the existing annotations.** Classification tasks are straightforward and well-defined, with strong classification performance often indicating robust feature learning. Therefore, they are suitable for evaluating video foundation models. We construct adaptation classification tasks based on the collected data and annotations as follow:

Table 2: **Task details of VideoEval.** All videos are collected from the public datasets for building tasks of VidTAB and VidEB.

| Domain | Task | Source | Task Description |
|---|---|---|---|
| *Video Task Adaptation Benchmark (VidTAB)* | | | |
| **Action Recognition in Special Scenarios** | Action Recognition in Dark Scene | ARID Xu et al. (2021b) | *Recognizing 11 distinct human actions in dark scenarios.* e.g. Run / Walk / Drink |
| | Action Recognition in Long Video | BreakFast Kuehne et al. (2014) | *Classifying 10 types of long-duration cooking videos.* e.g. Milk / Tea / Sandwich |
| **AI for Science** | Medical Surgery | SurgicalActions160 Schoeffmann et al. (2018) | *Classifying 16 surgical actions in gynecologic laparoscopy.* e.g. Knotting / Suction / Injection |
| | Animal Behavior | Animal Kingdom Ng et al. (2022) | *Classifying 12 behaviors of wild animals from diverse environmental footage.* e.g. Flying / Chirping / Preening |
| **Video Content Moderation** | Fake Face | FaceForensics++ Rossler et al. (2019) | *Determine whether the faces in the video have been tampered with by AI technology (such as DeepFake).* e.g. Origin video / Video with fake face |
| | Harmful Content | mob Ahmed et al. (2023) | *Detecting 3 degrees of malicious content within videos.* e.g. Obscene / Indecent activity / Violent activity |
| **Video Quality Assessment** | Quality Assess | DOVER Wu et al. (2023) | *Evaluating videos from an aesthetic and technical perspective and categorizing them into low and high quality.* e.g. Low quality / High quality |
| **Emotion Analysis** | Emotion Analysis | CAER Lee et al. (2019) | *Classifying 7 different human emotions in video.* e.g. Happy / Fear / Anger |
| *Video Embedding Benchmark (VidEB)* | | | |
| **Scene Understanding in Temporal Contexts** | Duplicate Scene Retrieval | FIVR5K Kordopatis-Zilos et al. (2019) | *Retrieve Duplicate Scene Videos (DSV):* Videos captured by the same camera and sharing at least one scene (without considering any application transformations). |
| | Complementary Scene Retrieval | FIVR5K Kordopatis-Zilos et al. (2019) | *Retrieve Complementary Scene Videos (CSV):* Retrieve a portion of the same spatiotemporal segment captured from different perspectives. |
| | Incident Scene Retrieval | FIVR5K Kordopatis-Zilos et al. (2019) | *Retrieving Incident Scene Videos (ISV):* The same event is close in both space and time, but there are no overlapping videos. |
| | Copy Detection | DVSC23 Pizzi et al. (2024) | *Detecting edited versions of the same source video.* Given a query inserted with one or more copied segments, detect the source video from the database. |

1. **Remove Low-Quality Video Datasets**: We manually exclude datasets with videos that have low resolution (below 240p), low frame rate (below 15fps), insufficient quantity (fewer than 150 videos per category), or low annotation accuracy (below 90%).

2. **Select Discriminative Tasks**: For task difficulty screening, we first evaluate zero-shot classification performance using CLIP-L Radford et al. (2021), EVA-g Sun et al. (2023), ViCLIP-L Wang et al. (2024a), and Internvideo2-1B Wang et al. (2024b). We then classify samples as follows: *Easy*: Samples that are correctly classified by three or more models. *Spatial*: Samples that are correctly classified by both CLIP and EVA. *Temporal*: Samples that are correctly classified by at least one of ViCLIP or Internvideo2-1B, but not by CLIP and EVA. *Hard*: Samples that are incorrectly classified by all models. We use the zero-shot classification accuracy of the models and the aforementioned proportions as references for task selection. Based on this, we choose tasks with lower zero-shot classification accuracy, higher proportions of Hard and Temporal samples, and lower proportions of Easy samples. The proportions of each type of sample in the tasks we ultimately selected can be found in Table 3.

3. **Control the Number of Categories**: For datasets that originally include category labels, such as ARID Xu et al. (2021b) and Animal Kingdom Ng et al. (2022), we select categories with sufficient samples to ensure evaluation accuracy and stability. We also control the final number of categories to avoid making the adaptation task overly difficult. We observed that both zero-shot testing and few-shot experiments based on current VFMs show that when the number of categories is too high, models often perform no better than random guessing. Although this issue may be mitigated as VFMs improve, we currently need to control the number of categories to effectively showcase differences between models. We select the main categories for each task and limit the number of categories to around 10 (based on few-shot experiments).

4. **Handling Multi-label and Regression Tasks**: For datasets that are not originally classification tasks, we transform the tasks into classification tasks. For example, for DOVER Wu et al. (2023), which is used for video aesthetics and technical quality assessment (a regression task), we assume that videos with quality scores in the top 40% are "high-quality videos" and those with scores in the bottom 40% are "low-quality videos", thus converting the original task into a binary classification task.

In total, we construct eight classification tasks to evaluate the adaptation capabilities of video foundation models.

**Determining the evaluation protocol.** Previous studies Wang et al. (2022b; 2024b); Yuan et al. (2023) typically train video models using entire samples of training set, and most popular benchmarks have large training sample sizes. We argue that this evaluation method overlooks the examination of the adaptation capability of VFMs. As illustrated in Figure 3, under the scenario of using full training samples, the differences between VFMs are difficult to discern. However, under a low-sample protocol, different foundation models exhibit varying degrees of task adaptation capabilities. We

Table 3: **Task difficulty assessment based on visual language models**. For tasks with fewer categories, such as Fake Face (n=2) and Quality Assess (n=2), random guessing can lead to high accuracy, which may result in a lower apparent proportion of hard samples. Therefore, the zero-shot classification accuracy of the models should also be considered when making task selection.

| ratio % | Dark Scene | Long Video | Medical Surgery | Animal Behavior | Fake Face | Harmfull Content | Quality Assess | Emotion Analysis |
|---------|-----------|-----------|-----------------|-----------------|-----------|------------------|----------------|------------------|
| Easy | 18.45 | 24.57 | 0.00 | 19.18 | 39.06 | 28.78 | 53.04 | 7.21 |
| Spatial | 19.00 | 20.44 | 4.17 | 20.86 | 20.72 | 24.56 | 51.24 | 5.01 |
| Temporal | 20.09 | 22.39 | 19.79 | 23.90 | 4.89 | 22.76 | 13.26 | 27.06 |
| Hard | 36.90 | 26.28 | 62.50 | 35.58 | 9.00 | 20.17 | 3.04 | 47.15 |

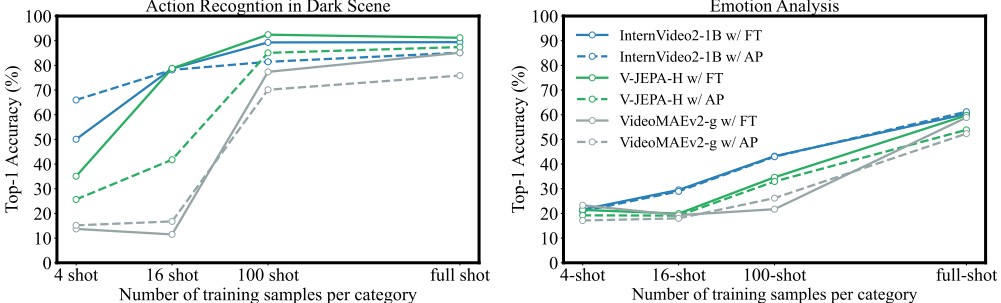

Figure 3: **Performance comparison on different training data scales.** We evaluate the performance variation of multiple video foundation models across tasks from two different domains as the scale of the training data changed. 'FT' and 'AP' denote full finetuning and attentive probe, respectively.

observe that for tasks such as Action Recognition in Dark Scenes, which VFMs usually excel at, there are significant differences in adaptation capabilities among different models when training samples are extremely limited (4 shot and 16 shot). As the number of samples gradually increases to 100 shot, these differences diminish. Conversely, for more challenging tasks like Emotion Analysis, the performances of different models are uniformly weak when training samples are extremely limited, showing no discernible differences until a certain number of training samples (100 shot) are reached, at which point different models begin to demonstrate distinct adaptation capabilities. Therefore, to account for the adaptation capabilities of models with different numbers of training samples, we define a task adaptation capability evaluation score (TA-score):

$$\text{TA-score} = \frac{Acc^{4s} + Acc^{16s} + Acc^{100s}}{3} \quad (1)$$

Where $Acc^{4s}$, $Acc^{16s}$, $Acc^{100s}$ represent the model's top-1 accuracy for 4-shot, 16-shot, and 100-shot classifications, respectively. Unless otherwise specified, we will use TA-score to denote the performance of various tasks in VidTAB.

Table 4: **Comparison of adaptation method on V-JEPA-H Bardes et al. (2023)** All results are obtained using A100-80G with PyTorch-builtin mixed precision, using a batch size of 4 to measure Cuda memory and training time. "Dark" and "Emotion" denote the tasks of Action Recognition in Dark Scenes and Emotion Analysis, respectively.

| Adaptation method | Tunable Params (M) | Cuda Memory (G) | Training Time (h) | Dark TA-score | Emotion TA-score |
|-------------------|--------------------|-----------------|-------------------|---------------|------------------|
| full finetuning | 663.7 | 52.1 | 1.0 | 68.8 | 25.3 |
| adapter | 52.6 | 45.0 | 1.0 | 62.4 | 24.7 |
| attentive probe | 19.7 | 6.4 | 0.4 | 54.7 | 23.8 |
| linear probe | 0.0 | 6.0 | 0.3 | 12.9 | 16.2 |

**Identifying efficient adaptation method for evaluation.** We also need to identify how to adapt the foundation models to the corresponding task. Previous work Houlsby et al. (2019); Yu et al. (2023); Pan et al. (2022); Yang et al. (2023); Li & Wang (2023) has explored various strategies for efficient adapting the foundation models. Here, we consider several of the most common and popular methods,

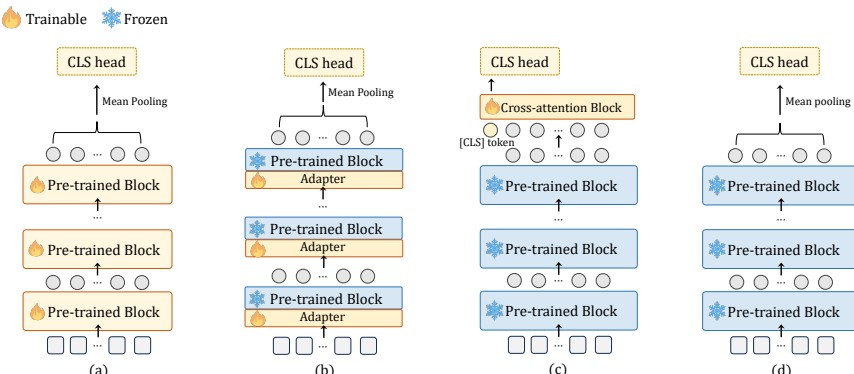

Figure 4: **Illustration of different adaptation method**: (a) Full Finetuning, (b) Adapter, (c) Attentive Probe and (d) Linear Probe.

as illustrated in Figure 4: **Full Finetuning**: Fine-tuning all the parameters of the pre-trained model. **Adapter**: Freezing the pre-trained model and inserting learnable low-rank adapter Pfeiffer et al. (2020) modules into each block of the pre-trained model for adaptation. **Attentive Probe**: Freezing the pre-trained model and adding an additional learnable cross-attention block at the end of the model to achieve attentive pooling, followed by a linear projection for classification. **Linear Probe**: Directly using the features from the pre-trained model, performing mean pooling, and then using a linear projection for classification. We evaluate the performance of these adaptation methods based on the V-JEPA-H model, as shown in Table 4. Full finetuning and adapter exhibited the best adaptation performance, but incurred high training costs. Linear probe was highly efficient but showed weak adaptation performance. Attentive probe offered a good trade-off between efficiency and adaptation performance. Therefore, in subsequent evaluation experiments, we employed attentive probe to adapt various vision foundation models.

## 3.2 VIDEO EMBEDDING BENCHMARK

The main application domains of video embeddings we considering include: Label-Level: Classification and Action Retrieval. Instance-Level: Retrieval, Copy Detection and Ranking. For label-level tasks, VidTAB has already provided a flexible way to evaluate models. Therefore, VidEB aims to assess existing models at a finer semantic level, focusing on instance-level tasks. Although ranking tasks are common in recommendation system scenarios, they are influenced by user information and interactions, in addition to video data. Based on prior research Ni et al. (2023), using frozen embeddings for video features does not consistently improve recommendation tasks (resulting in minimal or even negative effects). Thus, we have narrowed the final dataset scope to instance-level retrieval and copy detection. Apart from the traditional classification tasks, the evaluation of representations typically involves standard benchmarks such as video action retrieval Han et al. (2020a); Xu et al. (2019); Han et al. (2020b), which primarily rely on class labels. However, this approach often overlooks the overall scene context and exhibits an overlap with recognition tasks. In contrast, inspired by previous works Plummer et al. (2015); Pizzi et al. (2024); Wu et al. (2007); Jiang et al. (2014); Douze et al. (2021), we establish more rigorous criteria for embedding evaluation in Table 2. Specifically, we require the model to determine the priority and retrieve individual samples based on the overall similarity, rather than solely relying on class labels. This evaluation protocol provides a more comprehensive assessment of the model's capability to encapsulate subtle visual information.

**Evaluation protocol.** To facilitate fine-grained embedding evaluation, we incorporate two tasks for assessment: **(1) Hierarchical Video Retrieval** aims to retrieve videos from a database that closely matches the query video in terms of scene, viewpoint, and temporal context. According to previous work Kordopatis-Zilos et al. (2019), videos related to the query are categorized into three levels based on their similarity to the query: Duplicate Scene Videos (DSVs), Complementary Scene Videos (CSVs), and Incident Scene Videos (ISVs), as shown in Table 2: Consequently, the retrieval tasks are structured into three hierarchical levels:

- *Duplicate Scene Video Retrieval*: only DSVs are positive instances.
- *Complementary Scene Video Retrieval*: both DSVs and CSVs are positive instances.
- *Incident Scene Video Retrieval*: DSVs, CSVs, and ISVs are all positive instances.

For the evaluation metric, we follow Kordopatis-Zilos et al. (2019) to utilize the mean Average Precision (mAP) to assess the quality of video ranking. **(2) Video Copy Detection** aims to detect edited copies of the query video. Instead of the ranking/retrieval task where all video pairs need to be sorted according to video embedding similarity, it is required to identify a set of video pairs that contain edited versions of the given query. Following Pizzi et al. (2024), we consider the micro-AP ($\mu$AP) as our evaluation metric that operates on all queries jointly and takes the confidence scores into account.

## 4 BENCHMARKING VIDEO FOUNDATION MODELS

### 4.1 TARGETS AND DETAILS OF EVALUATION

**Evaluation targets**    We evaluate twenty open-source vision foundation models. Including: (1) twelve video foundation models, covering ***different pre-training paradigms, model scales, and training data scales***, to analyze the impact of these factors on the generalization capability of foundation models. (2) five image foundation models to observe ***how much generalization capability trained on image data can exhibit in video understanding***. (3) three image-to-video methods based on image foundation models to assess the ***effectiveness of current efficient transfer methods***.

**Implementation details**    All models take 8 frames (16 frames if the model has temporal downsampling), with each frame being 224x224 in size as input. For VidTAB, to ensure fair comparison and efficient assessment, we train all models for the same number of epochs and made minor adjustments to the hyperparameters to ensure convergence. For VidEB, all models take 16 frames, with each frame being 224x224 in size as input. In hierarchical video retrieval, the similarity of video-level embedding determines the ranking of retrieval results. In video copy detection, each sample is segmented into 5 clips. The detection confidence score for the entire video is derived from the maximum frame-wise similarity computed for each query-reference pair. See the Appendix for more details.

### 4.2 RESULTS ON VIDTAB

**Zero-shot evaluation**    To preliminarily assess the characteristics and difficulty of the dataset, we first evaluate the zero-shot performance of the eight tasks we created using two image language models and two video language models. As shown in the top section of Table 3, both image and video models demonstrated some level of performance for action-related tasks, with video models exhibiting relatively higher performance. For tasks involving low-level information understanding, such as Quality Assessment task, image models performed significantly better. In contrast, for other tasks involving scenarios typically unseen in training data, such as medical surgery videos or Safety Review tasks requiring complex semantic reasoning, all models exhibited almost no performance.

**Main results**    Table 5 presents the evaluation results on VidTAB. We summarize our findings as follows. **On the whole, (1)** Despite exhibiting a degree of generalization capability, *current vision FMs still struggle to adapt to unseen video tasks with limited training samples.* VFMs outperform IFMs, particularly in tasks related to action and behavior understanding. However, IFMs exhibit superior performance on more novel tasks, specifically in the domains of safety and quality, especially when combined with image-to-video adaptation techniques. **(2)** The *adaptation performance of models generally increases with the growth of data and model size*, as observed by the improvements observed from V-JEPA-L to V-JEPA-H (+1.5) and ViCLIP-L-10M to ViCLIP-L-200M (+1.3).

**For the pre-training data, (3)** *While augmenting video training data is generally beneficial, it can negatively impact the performance on some tasks.* For both VideoMAEv2-g and InternVideo2-1B$_{stage1}$, fine-tuning on Kinetics-710 data significantly enhances Action-related tasks, but consistently degrades certain Safety and Quality tasks. Similar findings are observed with ViCLIP-L, where post-pretraining on a large-scale video dataset improves Action-related tasks but diminishes performance in other domains (Science, Safety, Quality, Emotion). It could be attributed to the

Table 5: **Evaluating state-of-the-art FMs on the VidTAB**. The best and second-best results of foundation models are noted by blue and underline, respectively. 'I', 'V', and 'IV' denote image data, video data, and mixed image-video data, respectively. Data marked in gray indicates that the model uses a model trained on that data as initialization. 'K710ft' indicates that the model was fine-tuned with supervision using the labeled action recognition dataset Kinetics-710 (0.66M). Considering the random error in few-shot experiments, we conducted 3-fold experiments for both 4-shot and 16-shot settings, and used their mean as the final result. We also provide the results of full finetuning in the appendix.

| | # Params (M) | # Pretrain Data | Average | Action | | Science | | Safety | | Quality | Emotion |
|---|---|---|---|---|---|---|---|---|---|---|---|
| | | | | Dark Scene | Long Video | Medical Surgery | Animal Behavior | Harmful Content | Fake Face | Quality Assess | Emotion Analysis |
| Random | - | - | 22.7 | 9.1 | 10.0 | 6.3 | 8.3 | 33.3 | 50.0 | 50.0 | 14.3 |
| *Zero-shot performance of visual language models* | | | | | | | | | | | |
| CLIP-L Radford et al. (2021) | 428 | **I**-400M | 35.7 | 29.2 | 34.6 | 12.5 | 32.9 | 42.1 | 56.3 | 65.5 | 12.9 |
| EVA-CLIP-g Sun et al. (2023) | 1365 | **I**-2B | 36.0 | 32.8 | 37.2 | 9.4 | 28.5 | 39.6 | 52.8 | 69.5 | 17.9 |
| ViCLIP-L Wang et al. (2024a) | 428 | I-400M+**V**-200M | 33.6 | 26.2 | 37.5 | 8.3 | 29.3 | 32.1 | 52.2 | 53.9 | 29.0 |
| InternVideo2$_{stage2}$ Wang et al. (2024b) | 1350 | IV-1.1M+**IV**-25.5M | 40.6 | 37.1 | 40.2 | 11.5 | 45.2 | 59.1 | 51.3 | 56.1 | 24.3 |
| *Image Foundation Model* | | | | | | | | | | | |
| CLIP-L Radford et al. (2021) | 316 | **I**-400M | 43.2 | 31.9 | 37.8 | 32.3 | 37.4 | 54.2 | 58.2 | 66.6 | 27.6 |
| SigLiP-SO Zhai et al. (2023) | 444 | **I**-4.11B | 43.3 | 27.6 | 38.4 | 36.5 | 35.8 | 53.3 | 58.5 | 67.8 | 28.5 |
| EVA-g Fang et al. (2023) | 1035 | **I**-2B | 45.8 | 40.2 | 47.1 | 34.4 | 41.0 | 51.8 | 55.2 | 68.1 | 29.0 |
| DINOv2-L Oquab et al. (2023) | 317 | **I**-142M | 42.7 | 40.8 | 45.0 | 39.6 | 36.1 | 38.9 | 52.2 | 63.2 | 25.6 |
| DINOv2-g Oquab et al. (2023) | 1165 | **I**-142M | 44.4 | 37.8 | 46.4 | 42.7 | 36.0 | 48.5 | 53.2 | 64.3 | 26.3 |
| *Image Foundation Model with image-to-video adaptation method* | | | | | | | | | | | |
| ST-Adapter-CLIP-L Pan et al. (2022) | 328 | **I**-400M | 46.5 | 42.4 | 44.3 | 31.2 | 40.1 | 47.4 | 64.6 | 71.5 | 30.4 |
| AIM-CLIP-L Yang et al. (2023) | 328 | **I**-400M | 48.8 | 41.5 | 50.0 | 38.5 | 40.2 | 46.4 | 69.5 | 73.7 | 30.6 |
| ZeroI2V-CLIP-L Li & Wang (2023) | 303 | **I**-400M | 46.3 | 40.3 | 47.0 | 31.2 | 40.2 | 46.1 | 65.2 | 69.9 | 30.5 |
| *Video Foundation Model* | | | | | | | | | | | |
| ViCLIP-L-10M Wang et al. (2024a) | 316 | I-400M+**V**-10M | 41.8 | 31.2 | 42.7 | 30.2 | 35.3 | 47.9 | 53.9 | 66.2 | 26.9 |
| ViCLIP-L-200M Wang et al. (2024a) | 316 | I-400M+**V**-200M | 43.3 | 38.2 | 44.6 | 30.2 | 37.9 | 47.4 | 54.9 | 65.9 | 27.5 |
| VideoMAEv1-L Tong et al. (2022) | 316 | **V**-0.24M | 43.3 | 45.6 | 30.8 | 31.2 | 37.4 | 56.5 | 51.9 | 68.7 | 24.0 |
| VideoMAEv1-H Tong et al. (2022) | 651 | **V**-0.24M | 44.7 | 45.5 | 31.0 | 35.4 | 38.6 | 55.8 | 51.8 | 70.5 | 29.1 |
| VideoMAEv2-g Wang et al. (2023b) | 1037 | **V**-1.35M | 37.8 | 35.2 | 18.3 | 18.8 | 33.7 | 59.6 | 50.9 | 64.7 | 21.6 |
| VideoMAEv2-g$^{k710pt}$ Wang et al. (2023b) | 1037 | **V**-1.35M+K710ft | 54.0 | 76.4 | 72.6 | 50.0 | 42.4 | 43.8 | 56.9 | 63.2 | 27.0 |
| UMT-L$_{stage1}$ Li et al. (2023) | 316 | **V**-0.66M | 40.6 | 34.3 | 35.4 | 30.0 | 34.2 | 45.6 | 53.6 | 64.7 | 27.0 |
| UMT-L$_{stage2}$ Li et al. (2023) | 316 | V-0.66M+**IV**-25M | 44.0 | 34.2 | 43.9 | 22.9 | 39.4 | 63.9 | 53.0 | 67.3 | 27.4 |
| V-JEPA-L Bardes et al. (2023) | 318 | **V**-2M | 43.5 | 50.4 | 34.3 | 39.6 | 39.7 | 43.9 | 51.7 | 66.7 | 21.4 |
| V-JEPA-H Bardes et al. (2023) | 653 | **V**-2M | 45.1 | 53.8 | 37.6 | 35.4 | 40.4 | 47.3 | 53.0 | 68.1 | 25.1 |
| InternVideo2-1B$_{stage1}$ Wang et al. (2024b) | 1037 | **IV**-1.1M | 46.1 | 45.2 | 50.3 | 33.3 | 38.7 | 52.3 | 53.5 | 65.9 | 29.3 |
| InternVideo2-1B$_{stage1}$ Wang et al. (2024b) | 1037 | **IV**-1.1M+K710ft | 56.7 | 75.6 | 77.5 | 53.1 | 45.4 | 47.2 | 55.5 | 66.2 | 33.2 |
| InternVideo2-1B$_{stage2}$ Wang et al. (2024b) | 1037 | IV-1.1M+**IV**-25.5M | 53.6 | 66.0 | 71.1 | 38.5 | 50.0 | 53.6 | 54.7 | 64.3 | 30.3 |

limited diversity of the current video training data. **(4)** For models trained on single-modal visual data, *incorporating additional weak-supervised post-pretraining with visual-text data leads to significant improvements in adaptation capabilities*. This is evident in the performance gains observed from UMT-L$_{stage1}$ to UMT-L$_{stage2}$ (+3.6) and from InternVideo2-1B$_{stage1}$ to InternVideo2-1B$_{stage2}$ (+8.0). Interestingly, this finding contradicts previous conclusions drawn from commonly used action recognition benchmarks, suggesting that these benchmarks may introduce bias.

**For the pre-training paradigms of model, (5)** *The effectiveness of pre-training paradigms in scaling model size might not be adequately validated on popular action recognition benchmarks.* While VideoMAEv2 successfully scaled a model to 1B parameters using the dual masking strategy Wang et al. (2023b), its adaptation performance (37.7 vs 44.4) significantly declined compared to VideoMAEv1-H. Interestingly, VideoMAEv2-g demonstrated remarkable performance after fine-tuning on Kinetics-710 (0.66M), suggesting that the abundant labeled data may have compensated for the shortcomings of its pre-training performance. **(6)** *Single-modal self-supervised pre-training paradigms exhibit superior data efficiency compared to multimodal weakly-supervised pre-training paradigms*. Notably, V-JEPA and VideoMAEv1, trained solely on relatively small-scale unlabeled video data via self-supervised pre-training, demonstrate comparable or even superior performance to ViCLIP, which is trained on a massive dataset of video-text pairs.

**In addition, (7)** *Effective adaptation method for FMs is crucial.* Three image-to-video methods based on CLIP-L achieved significant performance improvements compared to using an attentive probe directly. We believe this represents a promising avenue for future research.

Table 6: **Evaluation of State-of-the-Art Foundation Models on the VidEB Dataset.** "K400pt" and "K400ft" denote that the model is pre-trained and fine-tuned, respectively, using the labeled action recognition dataset Kinetics-400 (0.31M). MCL: Multi-modal Contrastive Learning, SCL: Self-supervised Contrastive Learning, MVM: Masked Video Modeling, SFT: Supervised Fine-tuning. Other notations are consistent with those in Table 5.

| | Pretrain Tasks | # Pretrain Data | **Average** | Scene | | | |
| --- | --- | --- | --- | --- | --- | --- | --- |
| | | | | Duplicate | Complementary | Incident | Copyright |
| *Image Foundation Model* | | | | | | | |
| CLIP-L Radford et al. (2021) | MCL | **I**-400M | 43.0 | 41.1 | 46.4 | 52.0 | 32.3 |
| EVA-g Fang et al. (2023) | MCL | **I**-2B | 37.1 | 41.4 | 46.1 | 51.7 | 9.3 |
| SigLiP-SO Zhai et al. (2023) | MCL | **I**-4.11B | 38.6 | 40.6 | 45.5 | 51.5 | 16.9 |
| DINOv2-L Oquab et al. (2023) | SCL | **I**-142M | 45.6 | 49.0 | 53.5 | 54.3 | 25.6 |
| DINOv2-g Oquab et al. (2023) | SCL | **I**-142M | 48.6 | 50.5 | 55.1 | 56.0 | 32.8 |
| *Video Foundation Model* | | | | | | | |
| VideoMAEv1-L Tong et al. (2022) | MVM | K400pt | 12.9 | 14.5 | 15.1 | 13.2 | 8.8 |
| VideoMAEv1-L-K400ft Tong et al. (2022) | MVM+SFT | K400pt+ft | 27.4 | 27.6 | 30.2 | 30.3 | 21.6 |
| VideoMAEv2-g Wang et al. (2023b) | MVM | **V**-1.35M | 11.6 | 14.8 | 15.4 | 13.4 | 2.8 |
| VideoMAEv2-g-K710ft Wang et al. (2023b) | MVM+SFT | **V**-1.35M+K710ft | 37.4 | 33.8 | 37.1 | 37.1 | 41.7 |
| UMT-L$_{stage1}$ Li et al. (2023) | MVM | **V**-0.66M | 41.1 | 42.2 | 46.6 | 49.6 | 25.7 |
| UMT-L$_{stage1}$-K710ft Li et al. (2023) | MVM+SFT | **V**-0.66M+K710ft | 29.0 | 26.4 | 29.4 | 30.3 | 30.0 |
| UMT-L$_{stage2}$ Li et al. (2023) | MVM+MCL | V-0.66M+**IV**-25M | 34.2 | 33.4 | 37.3 | 40.6 | 25.4 |
| V-JEPA-L Bardes et al. (2023) | MVM | **V**-2M | 19.7 | 21.3 | 23.9 | 21.7 | 12.0 |
| V-JEPA-H Bardes et al. (2023) | MVM | **V**-2M | 20.2 | 21.5 | 23.7 | 21.2 | 14.3 |
| InternVideo2-1B$_{stage1}$ Wang et al. (2024b) | MVM | **IV**-1.1M | 50.4 | 47.3 | 52.1 | 54.9 | 47.3 |
| InternVideo2-1B$_{stage1}$-K710ft Wang et al. (2024b) | MVM+SFT | **IV**-1.1M+K710ft | 33.9 | 30.5 | 34.2 | 34.1 | 36.9 |
| InternVideo2-1B$_{stage2}$ Wang et al. (2024b) | MVM+MCL | IV-1.1M+**IV**-25.5M | 34.6 | 32.4 | 36.8 | 39.9 | 29.3 |

## 4.3 RESULTS ON VIDEB

The main results of VidEB are presented in Table 6. We evaluate the embedding performance using different pre-training paradigms for IFMs and VFMs as frozen feature extractors. Surprisingly, **IFMs performs better than most VFMs**, likely due to the existing gap in spatial modeling capabilities between VFMs and IFMs.

**For the pre-training paradigms of the model, (1)** *The contrastive learning (CL) based approach consistently excels in embedding evaluation.* Due to CL's emphasis on the relationships between samples during training, DINOv2, which focuses solely on vision, outperforms vision-language contrastive methods like CLIP across multiple tasks. **(2)** *The effectiveness of masked video modeling is closely tied to the targets it reconstructs or aligns with.* With higher semantic richness, it shows progressive improvements in embedding quality for VideoMAE-L, V-JEPA-L, and UMT-L$_{stage1}$. **(3)** *Vision-centric pretraining outperforms Multi-modal pretraining in vision-centric scenarios.* Comparing UMT-L$_{stage1}$ and InternVideo2-1B$_{stage1}$ with their multi-modal counterparts UMT-L$_{stage2}$ and InternVideo2-1B$_{stage2}$, the introduction of visual-text pair data in multi-stage training does not enhance performance in vision-centric scenarios. This is also consistent with the performance differences observed between DINO and CLIP-style pre-training methods.

Additionally, we assess the **impact of fine-tuning on the embedding evaluation of these pre-trained models**. **(4)** *Labels bring new semantic information or disrupt existing finer-grained semantic information.* The performance variations after fine-tuning differ based on the pre-training strategy. For UMT-L$_{stage1}$ and InternVideo2-1B$_{stage1}$, fine-tuning leads to a significant drop in performance (-12.1 for UMT and -16.5 for InternVideo) due to the introduction of more singular label information, which causes catastrophic forgetting. In contrast, VideoMAE and VideoMAEv2 show substantial performance gains (+14.5 and +25.8, respectively) because the low-level semantics learned during pre-training are less abstract and benefit more from the addition of high-level label information.

## 5 CONCLUSIONS

We present VideoEval, a comprehensive benchmark suite for efficiently evaluating the VFMs. To this end, we establish VidTAB, which explores suitable evaluation tasks and protocols for VFMs from the perspective of assessing their adaptability to unknown tasks with limited samples. Additionally, we create VidEB to evaluate the capability of VFMs' feature embedding in directly supporting downstream tasks. Utilizing VideoEval, we conduct a large-scale study involving 20 popular open-source vision foundation models, providing valuable insights for future research directions.

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

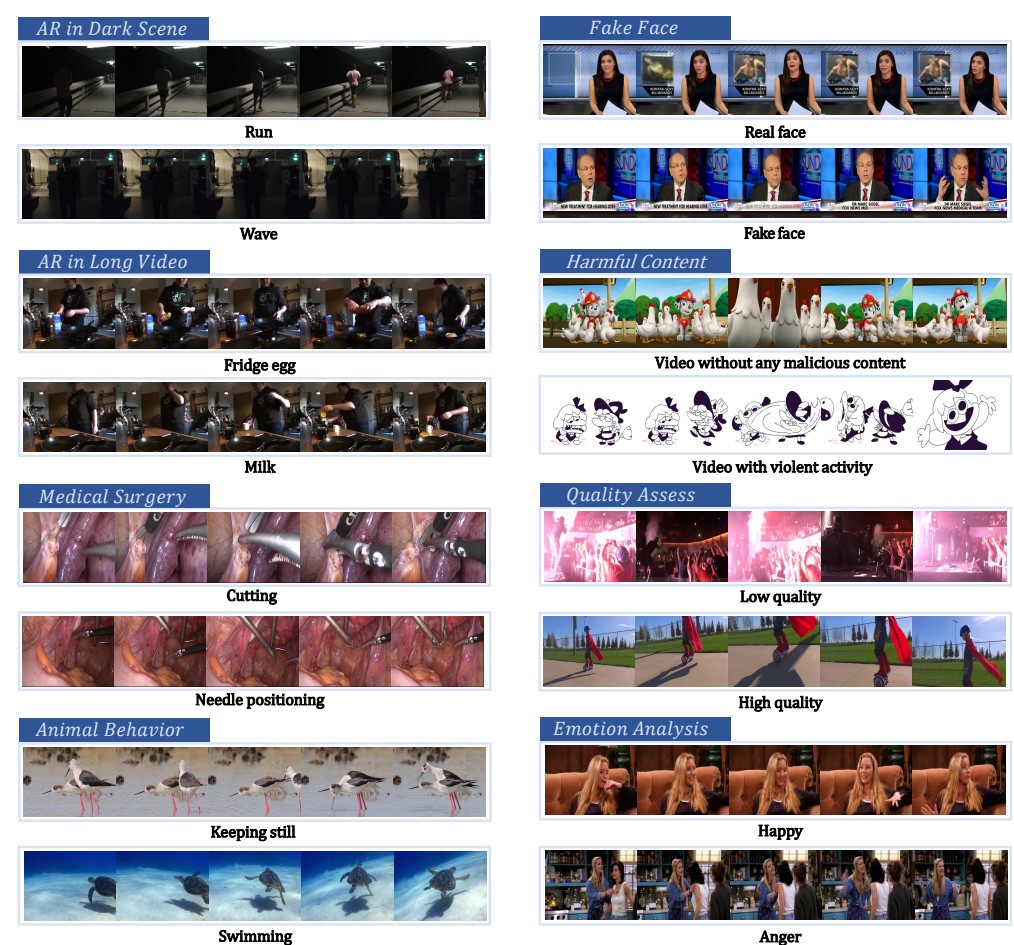

Figure 5: **Examples of VidTAB.**

In this appendix, we provide more details of VideoEval from the following aspects:

- Details of our benchmark are in § A.
- Details of training and evaluation, can be found in § B.
- Ethics etatement of the datasets are in § C
- Limitations and potential negative societal impacts are in § D

## A   DETAILS OF BENCHMARK

**Comparison of Current VFMs Benchmarks**   As shown in Table 1, we compare our VideoEval benchmark with existing benchmarks available for VFMs from the perspectives of evaluation cost and benchmark diversity.

**Examples of VidTAB**   As shown in Figure 5, we present some examples of tasks in VidTAB.

**Details of VidTAB**   The detals of task construction are presented in Table 7. For each category in one task, we sample 4, 16, and 100 samples, respectively. Given the limited volume of medical surgery data, we only sample 4 samples from each category for few-shot evaluation. To mitigate the impact of randomness, we sampled two sets of data for four tasks and obtained the benchmark results. We found that the randomness of sampling had negligible effects on the final rankings of VFMs in the benchmark.

Table 7: **Task details of VidTAB.** All videos are collected from the public datasets for building tasks of VidTAB.

| Task | Source | Num. test sample | Details of Task Construction |
|---|---|---|---|
| Action Recognition | ARID Xu et al. (2021b) | 2011 | We directly employ the original classification task definition. Specifically, 11 categories. |
| Action Recognition | BreakFast Kuehne et al. (2014) | 822 | We directly employ the original classification task definition. Specifically, 12 categories. |
| Medical Surgery | SurgicalActions160 Schoeffmann et al. (2018) | 96 | We directly employ the original classification task definition. Specifically, 16 categories. |
| Animal Behavior | Animal Kingdom Ng et al. (2022) | 2268 | Since the annotations in this dataset included multiple labels, we filtered out all categories with only single labels and then selected categories with more than 150 samples. This resulted in a final set of 12 categories. |
| Fake Face | FaceForensics++ Rossler et al. (2019) | 1800 | We used the original 1000 videos as positive samples. Then, we divided the original videos into five parts and used the Deepfakes, Face2Face, FaceShifter, FaceSwap, and NeuralTextures methods to generate 1000 negative samples by face-swapping. We then selected 1800 of these samples as the test set and the remaining as the training set. |
| Harmful Content | mob Ahmed et al. (2023) | 1661 | We categorized videos into three classes based on their content: those containing fast repetitive movements and violence activities, those containing unpleasant appearances and obscene scenes, and those containing no malicious information at all. This resulted in a three-class classification task. |
| Quality Assess | DOVER Wu et al. (2023) | 724 | To convert the task into a classification problem, we sorted the "overall score" label and divided the videos into positive and negative samples, with the top and bottom 40% constituting the respective categories. |
| Emotion Analysis | CAER Lee et al. (2019) | 3953 | We directly employ the original classification task definition. Specifically, 7 categories. |

## B DETAILS OF TRAINING AND EVALUATION

**Checkpoints of Evaluation Models** We provide checkpoints of the models we evaluate for reproducibility of our results.

- CLIP Radford et al. (2021): `https://huggingface.co/openai/clip-vit-large-patch14`
- EVA-CLIP Radford et al. (2021): `https://huggingface.co/QuanSun/EVA-CLIP`
- ViCLIP Wang et al. (2024a): `https://github.com/OpenGVLab/InternVideo/tree/main/Data/InternVid`
- InternVideo2 Wang et al. (2024b): `https://huggingface.co/collections/OpenGVLab/internvideo2-6618ccb574bd2f91410df5cd`
- SigLiP Zhai et al. (2023): `https://huggingface.co/google/siglip-so400m-patch14-384`
- DINOv2 Oquab et al. (2023): `https://huggingface.co/facebook/dinov2-giant`
- VideoMAE Tong et al. (2022): `https://github.com/MCG-NJU/VideoMAE/blob/main/MODEL_ZOO.md`
- VideoMAEv2 Wang et al. (2023b): `https://github.com/OpenGVLab/VideoMAEv2/blob/master/docs/MODEL_ZOO.md`
- UMT Li et al. (2023): `https://github.com/OpenGVLab/unmasked_teacher`
- V-JEPA Bardes et al. (2023): `https://github.com/facebookresearch/jepa`

**Trainging strategies** Specific hyperparameter configurations are available in the configs provided in our code repository. In essence, we train all models for 25 epochs using a similar training strategy, employing the Adam optimizer, a learning rate of 5e-5, and only utilizing RandomResizedCrop for data augmentation. And we use a single clip to obtain the final evaluation performance.

**Total amount of compute and the type of resources used** Leveraging the low cost of our evaluation protocol, we conducted each experiment involving a single VFM and a single task on one A100-80G GPU. We performed approximately 300 such experiments, each taking around 1-2 hours, resulting in a total of around 400 GPU hours.

## C ETHICS STATEMENT

**license of the datasets** The dataset we are using is collected from publicly accessible sources, all licensed under Creative Commons (CC-BY) or other open-source licenses. We have diligently followed all legal requirements to integrate this data into our research, emphasizing the importance of transparency in data licensing for proper attribution and appropriate use. Although we have taken steps to ensure the inclusion of suitable content, we recognize that some problematic content may still exist. If you encounter any such content, please notify us immediately so we can take corrective action to maintain a dataset free from inappropriate material. We are dedicated to maintaining a

Table 8: **Evaluating state-of-the-art VFMs on the VidTAB with Full Finetuning**. The best and second-best results of foundation models are noted by blue and underline, respectively. We present the results in the form of '4s/16s/100s,' representing the outcomes of 4-shot, 16-shot, and 100-shot experiments.

| | | Action | | Science | | Safety | | Quality | Emotion |
| | | Dark Scene | Long Video | Medical Surgery | Animal Behavior | Harmful Content | Fake Face | Quality Assess | Emotion Analysis |
| | Average | | | | | | | | |
|---|---|---|---|---|---|---|---|---|---|
| Random | 22.7 | 9.1 | 10.0 | 6.2 | 8.3 | 33.3 | 50.0 | 50.0 | 14.3 |
| *Video Foundation Model* | | | | | | | | | |
| ViCLIP-L-10M Wang et al. (2024a) | 37.9 | 22.6/18.9/29.5 | 16.4/24.8/45.7 | 30.2 | 26.3/29.7/41.5 | 35.1/38.2/54.2 | 51.2/50.8/53.7 | 56.9/65.9/72.5 | 20.3/17.2/32.7 |
| ViCLIP-L-200M Wang et al. (2024a) | 38.3 | 21.1/20.5/37.2 | 13.6/21.2/53.0 | 30.2 | 25.1/30.6/43.6 | 36.6/40.2/46.8 | 50.4/51.5/53.7 | 57.2/67.7/71.6 | 19.8/19.7/32.2 |
| VideoMAEv1-H Tong et al. (2022) | 34.0 | 12.8/13.5/72.1 | 9.6/10.0/36.7 | 39.6 | 18.5/22.0/47.8 | 32.5/33.1/37.2 | 50.3/50.3/50.7 | 44.2/50.8/66.6 | 15.2/14.3/19.0 |
| VideoMAEv2-g Wang et al. (2023b) | 34.0 | 13.1/13.4/76.1 | 31.4/12.3/34.3 | 18.8 | 12.2/18.7/50.8 | 29.4/30.2/41.5 | 50.8/50.6/50.6 | 52.0/55.3/62.2 | 12.7/14.2/17.4 |
| VideoMAEv2-g$^{k710pt}$ Wang et al. (2023b) | 48.6 | 30.4/77.3/ **94.0** | 31.2/52.9/89.0 | 57.3 | 12.6/32.0/64.5 | 33.1/39.4/41.8 | 49.8/50.4/54.7 | 54.3/59.8/71.4 | 16.6/17.2/39.3 |
| V-JEPA-L Bardes et al. (2023) | 49.2 | 43.2/78.8/88.5 | 25.2/52.0/86.0 | 46.9 | 26.6/37.1/59.9 | 38.5/36.0/46.4 | 50.2/50.8/55.9 | 54.3/68.0/76.9 | 15.0/17.9/27.4 |
| V-JEPA-H Bardes et al. (2023) | 52.5 | 45.2/ **80.7** /90.8 | 24.7/48.5/87.1 | 46.9 | 26.7/38.1/60.6 | 40.4/41.7/ **58.5** | 50.4/51.2/68.2 | 59.8/ **71.3** / **79.3** | 20.9/20.4/43.4 |
| InternVideo2-1B$_{stage1}$ Wang et al. (2024b) | 52.1 | 20.3/56.0/80.6 | 27.7/70.0/92.5 | 66.7 | 27.2/38.2/58.8 | 41.5/36.0/50.0 | 52.6/52.4/75.0 | **60.9** /69.0/77.8 | 16.1/31.8/45.4 |
| InternVideo2-1B$_{stage1}^{k710pt}$ Wang et al. (2024b) | **59.4** | 59.5/79.9/88.9 | **60.8 / 82.6 / 95.6** | 71.9 | 31.7/46.4/ **68.0** | 44.0/37.7/50.1 | 53.3 / 53.9 / 83.2 | 59.4/65.4/77.9 | 22.9/28.0/ **45.8** |
| InternVideo2-1B$_{stage2}$ Wang et al. (2024b) | 59.0 | 55.1/75.6/89.3 | 55.4/77.7/93.7 | 60.4 | 33.3 / 51.0 /67.7 | 54.2 / 42.2 /55.1 | 50.9/53.4/76.9 | 58.5/67.0/77.4 | 23.9 / 34.6 /44.1 |

high-quality, ethically responsible dataset and pledge to uphold principles of privacy and transparency in all our work.

**Privacy or safety concerns in video** For personally identifiable information or offensive content in video, our data collection sources have been carefully considered, and we believe these issues are not present. However, if you discover any oversights, please do not hesitate to contact us promptly.

# D LIMTIATIONS AND SOCIETAL IMPACTS

**Limitations** Firstly, due to the limitations of diversity and accuracy in our video sources and annotations, which were gathered from public resources, we plan to further enrich the task in the future by incorporating manual annotations and self-collected data. Secondly, considering the evaluation cost and simplicity, we currently only evaluate tasks like classification and retrieval, which primarily rely on VFMs' global information extraction capabilities. We have not yet considered tasks like spatio-temporal action detection and temporal grounding, which assess other aspects of VFMs' capabilities. We will expand the scope of evaluation in the future.

**Potential negative societal impacts** While our evaluation includes tasks like synthetic video recognition and harmful information recognition, these serve only as indicators of the model's overall performance in this area and cannot be used to accurately evaluate the actual performance of a specific task. If researchers or engineers in society attempt to use VFMs to perform these specific tasks, our benchmark can serve as a reference for their choice of VFMs, but it cannot be used as the final standard for evaluating that task. Otherwise, it may have negative impacts on the corresponding real-world applications.

Table 9: **Evaluating state-of-the-art FMs on the VidTAB with Attentive Probe**. The best and second-best results of foundation models are noted by  blue  and underline, respectively. We present the results in the form of '4s/16s/100s,' representing the outcomes of 4-shot, 16-shot, and 100-shot experiments.

| | Average | Action | | Science | | Safety | | Quality | Emotion |
| | | Dark Scene | Long Video | Medical Surgery | Animal Behavior | Harmful Content | Fake Face | Quality Assess | Emotion Analysis |
|---|---|---|---|---|---|---|---|---|---|
| Random | 22.7 | 9.1 | 10.0 | 6.2 | 8.3 | 33.3 | 50.0 | 50.0 | 14.3 |
| *Image Foundation Model* | | | | | | | | | |
| CLIP-L Radford et al. (2021) | 44.3 | 20.5/21.6/53.5 | 15.1/21.8/76.6 | 32.3 | 29.5/36.3/46.4 | 49.5/48.1/65.1 | 52.8/57.1/64.6 | 60.2/69.4/70.3 | 21.8/22.8/38.2 |
| SigLiP-SO Zhai et al. (2023) | 43.9 | 20.1/23.7/39.0 | 16.8/27.5/71.0 | 36.5 | 25.0/35.0/47.4 | 49.8/48.1/62.1 | 54.8/57.2/63.4 | 58.8/68.9/75.7 | 21.7/23.4/40.5 |
| EVA-g Fang et al. (2023) | 46.9 | 26.8/33.5/60.3 | 22.1/36.7/82.5 | 34.4 | 31.9/39.9/51.3 | 49.6/45.5/60.4 | 51.6/55.1/58.8 | 60.4/69.3/74.6 | 23.2/24.2/39.7 |
| DINOv2-L Oquab et al. (2023) | 42.9 | 26.2/37.3/58.9 | 17.0/37.1/80.8 | 39.6 | 26.5/36.3/45.4 | 37.1/31.6/48.0 | 51.0/52.0/53.6 | 54.7/64.5/70.3 | 21.8/22.5/32.4 |
| DINOv2-g Oquab et al. (2023) | 44.5 | 23.7/33.7/56.1 | 17.7/38.4/83.2 | 42.7 | 26.6/36.1/45.4 | 40.9/44.2/60.3 | 51.8/51.8/55.9 | 54.5/65.5/72.8 | 21.8/22.7/34.4 |
| *Image Foundation Model with image-to-video adaptation method* | | | | | | | | | |
| ST-Adapter-CLIP-L Pan et al. (2022) | 47.9 | 21.2/37.3/68.6 | 17.4/35.1/80.5 | 31.2 | 30.1/39.6/50.7 | 48.0/42.9/51.4 | 53.3/59.6/80.8 | 62.4/71.9/80.1 | 20.3/22.1/48.7 |
| AIM-CLIP-L Yang et al. (2023) | 49.7 | 22.4/39.3/62.6 | 21.2/47.5/81.3 | 38.5 | 29.7/39.1/51.7 | 44.3/38.9/55.9 | **57.4 / 67.2 / 83.8** | **64.9 / 73.0 / 83.2** | 21.8/24.7/45.2 |
| ZeroI2V-CLIP-L Li & Wang (2023) | 47.6 | 22.2/37.8/61.0 | 21.2/40.6/79.1 | 31.2 | 31.0/39.4/50.1 | 40.9/37.9/59.5 | 55.5/57.7/82.3 | 58.8/70.4/80.4 | 20.1/22.6/ **48.7** |
| *Video Foundation Model* | | | | | | | | | |
| ViCLIP-L-10M Wang et al. (2024a) | 42.8 | 22.4/25.2/46.1 | 19.6/35.3/73.2 | 30.2 | 26.3/34.4/45.2 | 38.0/46.9/58.8 | 51.6/53.4/56.8 | 59.5/68.0/71.0 | 21.2/22.4/37.1 |
| ViCLIP-L-200M Wang et al. (2024a) | 44.5 | 25.9/32.4/56.2 | 21.1/38.0/74.7 | 30.2 | 28.2/37.0/48.6 | 45.8/44.6/51.8 | 52.5/53.6/58.7 | 56.4/70.2/71.1 | 21.0/23.2/38.2 |
| VideoMAEv1-L Tong et al. (2022) | 44.4 | 19.0/35.1/82.6 | 12.8/14.6/65.1 | 31.2 | 25.6/31.1/55.4 | 62.1/49.6/57.9 | 50.5/51.1/54.2 | 57.9/70.3/77.9 | 18.9/16.7/36.5 |
| VideoMAEv1-H Tong et al. (2022) | 45.6 | 17.7/35.4/83.4 | 11.8/15.7/65.6 | 35.4 | 24.8/32.8/58.2 | 56.0/45.6/ **65.9** | 50.4/51.3/53.6 | 62.6/70.4/78.4 | **26.1** /26.4/34.8 |
| VideoMAEv2-g Wang et al. (2023b) | 39.6 | 15.9/19.6/70.0 | 14.2/14.0/26.8 | 18.8 | 25.2/26.1/49.7 | 63.1/52.9/62.7 | 50.9/50.5/51.2 | 56.5/62.6/74.9 | 16.7/21.9/26.2 |
| VideoMAEv2-g$^{k710pt}$ Wang et al. (2023b) | 54.4 | 63.4/76.9/ **88.8** | 59.5/75.3/83.0 | 50.0 | 26.5/41.3/59.3 | 41.0/41.4/49.1 | 52.9/55.2/62.6 | 52.3/65.3/72.1 | 21.4/23.1/36.6 |
| UMT-L$_{stage1}$ Li et al. (2023) | 41.6 | 25.5/21.8/55.5 | 14.8/22.4/68.9 | 30.0 | 24.9/32.8/44.8 | 42.4/41.4/53.0 | 51.1/52.9/56.9 | 59.3/66.3/68.5 | 24.2/20.0/36.9 |
| UMT-L$_{stage2}$ Li et al. (2023) | 45.9 | 25.2/26.6/50.8 | 23.6/35.2/72.8 | 22.9 | | **66.6 / 61.8** /63.3 | 50.6/51.4/56.9 | 58.9/68.5/74.4 | **25.0** /20.5/36.6 |
| V-JEPA-L Bardes et al. (2023) | 43.8 | 26.8/46.7/77.8 | 18.1/27.5/57.4 | 39.6 | 28.0/36.0/55.2 | 37.1/41.3/53.2 | 50.9/50.9/53.4 | 55.2/67.6/77.2 | 18.5/17.8/27.8 |
| V-JEPA-H Bardes et al. (2023) | 46.0 | 28.1/47.5/ _85.7_ | 17.2/26.9/68.6 | 35.4 | 27.6/36.6/57.0 | 40.4/42.0/59.6 | 51.3/52.5/55.3 | 58.0/68.4/77.9 | 22.1/20.3/32.9 |
| InternVideo2-1B$_{stage1}$ Wang et al. (2024b) | 47.2 | 27.4/38.5/69.7 | 22.2/42.5/ _86.1_ | 33.3 | 28.5/36.3/51.3 | 44.7/48.2/64.1 | 50.8/53.0/56.8 | 57.6/67.1/73.1 | 23.0/24.0/40.9 |
| InternVideo2-1B$_{stage1}$ Wang et al. (2024b) | **57.0** | **66.4 / 77.9** /82.4 | **65.3 / 77.5 / 89.8** | **53.1** | 31.3/ _44.1_ /60.7 | 43.9/42.4/55.4 | 51.9/54.7/59.9 | 57.1/66.5/75.0 | 23.3/ **33.3** /43.0 |
| InternVideo2-1B$_{stage2}$ Wang et al. (2024b) | _54.9_ | 54.4/66.6/76.9 | 56.0/71.7/85.6 | 38.5 | **37.2 / 50.4 / 62.5** | 51.0/46.2/63.6 | 51.6/54.4/58.2 | 53.9/65.8/73.2 | 21.8/ _29.3_ /39.9 |

