# OpenReview forum: "VideoEval: Comprehensive Benchmark Suite for Low-Cost Evaluation of Video Foundation Model"
_ICLR.cc/2025/Conference — ICLR 2025 Conference Withdrawn Submission_

### Official Review · Reviewer_H4dz · 2024-10-21

**Soundness:** 3
**Presentation:** 3
**Contribution:** 2
**Rating:** 5
**Confidence:** 4

**Summary:**

Authors aim to address the evaluation of video foundation models via a comprehensive benchmark. The author introduces VideoEval to address the limitations of traditional benchmarks (and evaluation protocols), namely poor diversity, high evaluation cost, and saturated performance metrics for Video Foundation Models. Work is partitioned into (a) VidTAB (few-shot learning) and (b) VidEB (feature embedding’s direct applicability to downstream tasks). Work evaluates 20 Foundation models, revealing:
(a) weak generalization across diverse tasks
(b) dataset size doesn’t increase the performance
(c) effectiveness of pretraining is understudied
(d) combining different training pretraining paradigms helps models generalize well.

**Strengths:**

(1)	**Evaluating / inferencing based on 20 models** strengthens the results.


(2)	**Variety of tasks**, namely: action recognition, “AI for science”, “video content moderation”, “video quality/aesthetic assess” and “emotion analysis” further strengthens the findings.


(3)	Work **addresses the shortcoming of conventional benchmarks**:
(a)	Expanding diversity of evaluation (contrary to traditional benchmarks which mainly focus on action recognition) (8 classification tasks)
(b)	The Diversity of the evaluation set and evaluation protocols differentiates between models, despite their saturation in performance on traditional datasets. The idea of a "TA-score" is really good and well-motivated.
(c)	Conventional evaluation often necessitates end-to-end training (not practical for Large VFMs), thus using few-shot learning as a low-cost alternative.
(d)	Vision-centric, focusing on vision models, discounting performance differences that may be caused because of language models.


(4)	**Paper is well-written** and easy to understand


(5)	VidTAB (few-shot learning with 8 tasks) & VidEB (embedding evaluation on downstream 4 tasks) both tasks provide a good comparison for models.


(6)	Some insights namely: “augmenting video training data can sometimes negatively affect certain tasks”, and the effect of “pre-training paradigms” are interesting.

**Weaknesses:**

1. **Clarity Needed**: How are authors defining “Beyond Action recognition” “Task Diversity” and “Domain Diversity”, e.g. VideoGLUE Yuan et al. (2023) is restricted to action recognition while it is task and domain-diverse. (Table 1).

2. **Comparison with previous work is misleading**:
a.	**VideoGLUE** (Yuan et al. (2023)) is restricted to action recognition only, although it consists of other tasks like temporal localization, and spatiotemporal localization, while this work counts “Medical Surgery” (medication action recognition) as separate task from normal action recognition.
b.	**BEAR** (Deng et al. (2023)) is counted as only action recognition while it does  “anomaly (violence) classification” which this work counts as separate content moderation. Similarly, their “Instructional Video” classification is counted as a separate task as AI for Science via “Medical Surgery”. Their “Gesture” recognition is similar to “Emotion Analysis” and “Animal Behavior” which are counted as separate tasks in this work.
c.	**VidTAB is classification only** (albeit different datasets). Temporal action localization is a much harder and a diverse task compared to classification. [Line 87-88] “Focus on action recognition, overlook other video understanding scenarios” This motivation is weak. Maybe reword this?

3. **Figures are difficult to understand**: Figure 1 can be easily summarized and it is occupying too much space. The same applies for Figure 2. Comic San font is hard to read and difficult to differentiate what part of the figure is important and what part belongs to the setup environment. Generally, these fonts denote dataset specific / setup, while methodology / proposed technique is in default font (but its opposite here?). Table 2 (and Table 3) are unreadable. So small font size. Put it in supplementary and expand it?

4. **Choice of words ‘Evaluation’ vs ‘Comparison’**: [Line 020-021] “*evaluating* the task adaptability of VFMs under few-shot conditions”. Benchmarks traditionally evaluate the models on a variety of tasks. Comparison among models is the byproduct of these benchmarks.
(a)[line 232-233] “exclude datasets with videos that have low resolution”,
(b) [line 250-251] “the number of categories is too high, models often perform no better than random guessing”.
These preprocesses are mainly done for model comparison and not in the spirit of true evaluation.

5. **Authors seemed to have missed the technique of “Visual Prompt Tuning”** (Jia, Menglin, et al. "Visual prompt tuning." European Conference on Computer Vision. Cham: Springer Nature Switzerland, 2022.) for “Identifying efficient adaptation method for evaluation”? It just adds 50 spatial tokens to adapt to models for various fine-tuning tasks.


6. **I’m doubtful if [Line 107] Low-cost is truly a contribution**. This study deals with foundation models which are especially good in zero-shot evaluation. It's unfair to count it as a contribution when dealing with such models, and comparing traditional benchmarks with traditional models. Additionally, What’s “’K710ft” Kinetics-710 (0.66M) is this few shot? 0.66M is not low cost as indicated in Figure 1.


7. **Space allocation is unconventional**. For a benchmarking paper, the setup for benchmarking takes the majority of space, while the benchmarking analysis starts at the 8th page (out of 10 pages) with minimal analysis. It's not well-reasoned why image models are included when benchmarking video models.


8. **[LINE 424-425] “The adaptation performance of models generally increases with the growth of data and model size” This is not necessarily true and requires a deeper analysis**.
(a).	Within similar models (i) “VideoMAEv1-H” has 651 M parameters with superior performance to “VideoMAEv2-g” (1037M). (ii) “ViCLIP-L-200M” (200 M pretraining) is inferior to ViCLIP-L-10M (10 M pretraining) on “Harmful Content”, and “Quality Assess”.
(b).	Outside the model the number of contractions is far greater. E.g. VideoMAEv2-g < UMT-Lstage1.

**Questions:**

Please address all the weakness
Also please indicate: Table 5 what models are fine-tuned and what are zero-shot.
How are authors justifying their contribution (d) Vision Centric, line 111 "avoiding the introduction of biases that may arise from incorporating language models"

---

### Official Review · Reviewer_MzWr · 2024-10-23

**Soundness:** 2
**Presentation:** 4
**Contribution:** 3
**Rating:** 5
**Confidence:** 4

**Summary:**

The paper introduces a benchmark for the evaluation of video foundation models (VFM). The benchmark is carefully curated based on a selection of existing video datasets. It is split into two sets: (1) a task adaptation benchmark (Video-TAB) that tests model performance on a variety of video tasks after lightweight adapter-based fine-tuning, and (2) an embedding benchmark (VidEB) that tests the performance of video embeddings, extracted using the benchmarked foundation models, on retrieval-related video tasks. Based on these benchmarks, the paper provides a survey of 20 current VFMs and offers insights into their applicability to different tasks. Furthermore, it discusses the effect of training data and training recipes on model performance.

**Strengths:**

Originality:

* The paper performs a large-scale evaluation of video foundation models on a new benchmark consisting of a diverse set of tasks. While this is not the only such survey, I am not aware of any studies that evaluate such a large amount of VFMs on such a large diversity of tasks, so I see the main novelty in the sheer breadth of the evaluation and the conclusions drawn from the state of the art in video foundation models.

Quality:

* The benchmark was carefully curated to contain high quality, challenging video datasets with good diversity.
* The paper evaluates 20 different models, including image-only baselines, image models adopted to video and video foundation models.
* The benchmark suite is comprehensive and diverse, consisting of 6 different domains and 12 different tasks and thus offers a good overview of VFM applicability to different tasks
* Design / scope choices made are clearly motivated and backed up with experiments, e.g. focus on few-shot setting, using attentive probe for model adaptation.

Clarity:

* The paper is well-organized and well-written.

Significance:

* Both adaptation and retrieval with video embeddings are evaluated
* The paper draws several interesting conclusions from their findings that point out weaknesses and directions for further research.

**Weaknesses:**

It seems to me that there is a disconnect between the discussion of the results and the actual results. Conclusions drawn are not supported by the data while some clear trends in the data are not discussed. While I would not recommend this paper for acceptance in its current state, I would be in favor of accepting it if the weaknesses above can be addressed sufficiently.

* The main weakness of the paper is that many of the conclusions it draws are not fully supported by the experimental results.
  * On VidTAB:
    * l. 421: “current vision FMs struggle to adapt to unseen video tasks”. What is the evidence for this statement? A comparison of VFM performance to SotA on these datasets would help support this.
    * l. 422: “VFMs outperform IFMs”: The best IFM beats 9 out of the 12 evaluated VFMs, so this statement doesn’t seem to be generally true, and also not in action and behavior tasks where some IFMs show stronger performance than most VFMs. So this statement should be weakened and results discussed in more nuance. This is actually an interesting negative result that warrants further discussion.
    * l. 431: “Similar findings are observed with ViCLIP-L, where post-pretraining on a large-scale video dataset improves Action-related tasks but diminishes performance in other domains (Science, Safety, Quality, Emotion)”: This statement is not supported by results in Tab. 5: VICLIP-L performance actually increases on science tasks, only slightly drops on one safety task while improving on the other, and increases on emotion.
  * On VidEB:
    * Conclusion (1): “The contrastive learning (CL) based approach consistently excels in embedding evaluation.”: This does not seem to be the case as two MVM methods perform better than the best CL method.
    * Conclusion (2): “The effectiveness of masked video modeling is closely tied to the targets it reconstructs or aligns with.”: This statement is hard to understand without context about the training data. What are the targets? How is “higher semantic richness” introduced?
* Results also suggest conclusions that are not discussed in the paper. These should be addressed and discussed in more detail:
  * VidTAB: A missing insight is that VFMs show no gain over IFMs on spatial tasks (Safety and Quality).
  * Image-to-video methods do better than VFMs on most tasks.
* From the evaluation results on both benchmarks it is not clear what the gap between VFMs and state of the art on each dataset is since results on specialist models on each task are not given.
* There are significant differences in results between the VidTAB and VidEB benchmarks that are not explained.
  * Why does stage 2 training improve UMT-L and InternVideo2 performance on VidTAB but degrades performance on VidEB?
  * Similarly, why does fine-tuning on K710 increase performance on VidTAB but hurts performance on VidEB?
* The related work section does not discuss image foundation models and image-to-video adapter approaches. It would be good to at least discuss those approaches that are evaluated. This would allow readers who are less familiar with the literature in this area to better interpret the results.
* Evaluation of task adaptation is limited to one method (attentive probe).
* Future research directions are not clearly pointed out, though some ideas are given in the results discussion. A separate section on this could help direct future research.

**Questions:**

* Will this benchmark be publicly available? If so, this would be a major contribution and should be mentioned in the paper.
* Tab. 4 motivates the choice to use attentive probe as the adapter method of choice for the evaluation by showing its good cost-performace tradeoff. However, results are only shown on V-JEPA-H. Do the findings hold true for other methods?
* As a reader unfamiliar with the attentive probe adapter, I was not able to understand how it works from just Fig. 4\. Could authors provide literature references and maybe expand the explanation in Sec. 3.1?
* Sec. 3.1: What other tasks / datasets were considered that did not end up in the benchmark? What were the reasons for them being excluded?
* In Tab. 5 it would be interesting to see which models perform well at spatial vs. temporal tasks. Have you considered dividing the evaluation results into the categories from Tab. 3?
* Fig. 3 motivates the choice of averaging three few-shot settings for scoring by showing that few-to-medium shot settings better separate the model accuracies. However this is only shown on two tasks. Do the other tasks exhibit the same separation properties?
* Sec. 4.1: How are the 8 / 16 frames that are fed into the models sampled from the videos? Does the evaluation use multiple temporal crops?
* Sec. 4.1: How are image models applied to video?
* I do not understand conclusion (5) in Sec. 4.2. “The effectiveness of pre-training paradigms in scaling model size might not be adequately validated on popular action recognition benchmarks.” What benchmarks is scaling not validated on and what are the results? What does “adaptation performance” mean? Could the authors clarify the meaning of this statement and explain how the following discussion supports it?
* Some numbers in Sec. 4.2 do not match the table, e.g. “+3.6” in l 470, 8.0 in l. 471 and 37.7, 44.4 in l. 476\.
* Sec. 4.3: How are videos ranked when using image models?
* l. 521: “This is also consistent with the performance differences observed between DINO and CLIP-style pre-training methods.” Which results does this statement refer to?
* Sec. 4/3, conclusion 4: “Labels bring new semantic information or disrupt existing finer-grained semantic information.” Could authors provide more context on the pre-training data used for the models discussed in this section? The conclusions are hard to understand without this context.
* l. 471: “previous conclusions” should be cited.
* l. 484: “Effective adaptation method for FMs is crucial.” This conclusion is not clear to me. Was the attentive probe not used for image-to-video methods? If not, which results are they being compared to?

Minor points:

* It’s often difficult to map author names of cited approaches to the actual names of the approaches. For citations that propose an approach with a well-known name, I’d suggest providing the names along with the citation, e.g. “V-JEPA (Bardes et al., 2023)”. That would also make it easier to map the methods mentioned in the related work section to Tab. 5\.
* Citations should have parentheses around them, otherwise it looks like they are part of the sentence. Please use “\\citep” to achieve this.
* The name VideoEval is very generic and would apply to any video benchmark. I’d suggest choosing a more specific name, e.g. “VideoFMEval”
* Minor grammatical error in the title: Models should be plural, so the title should be “\[...\] evaluation of video foundation model**s**”
* The font size of Tab. 2 is a little small and the table is hard to read in print.
* The radar chart in Fig. 1 (top right) is very interesting but hard to read at this size. I’d suggest enlarging it or even moving it into its own figure.
* Please right-justify numeric columns in tables.
* l. 413: The table reference should be changed to Tab. 5\.
* Could authors better explain the terms DSVs, CSVs and ISVs in Sec. 3.2?

---

### Official Review · Reviewer_jNwF · 2024-10-27

**Soundness:** 2
**Presentation:** 3
**Contribution:** 1
**Rating:** 3
**Confidence:** 5

**Summary:**

The paper presents a benchmark dataset for the evaluation of video foundation models. It is focused on video classification (instead of video -to-text, video-question-answering, etc.). All the source videos in the dataset are selected from existing public datasets. The paper reported results for a number of existing video foundation models.

**Strengths:**

The dataset is useful for researchers to evaluate their video encoders.

**Weaknesses:**

The paper did not evaluate any of the large multimodality models such as GPT-4V, Gemini, video-llava, etc. It seems that the paper regards "video foundation models" as "video encoders", and the benchmark can only be used to evaluate video encoders.

The paper focuses on classification tasks. Modern video understanding tasks have shifted from classification to more fine-grained understanding such as (dense) text description, video question answering, etc.

Only a few (8 vidTab, 16 vidEB) frames are selected from each video. The benchmark won't be able to evaluate long video understanding capabilities, and it won't be able to capture the speed of motion.

**Questions:**

Are video-language models regarded as video foundation models?

Can the benchmark be used to evaluate video-language models such as GPT-4V, video-llava, etc?

Does the benchmark cover surveillance monitoring scenarios where there may be multiple people in the scene and each person performs a different action?

Does the benchmark cover the capability to evaluate how well a person performs certain actions?

---

### Official Review · Reviewer_xHyA · 2024-10-30

**Soundness:** 3
**Presentation:** 2
**Contribution:** 2
**Rating:** 3
**Confidence:** 4

**Summary:**

The paper proposes an ensemble of video datasets for foundation model benchmarking, and provides different evaluation protocols. The proposed benchmarks consist of two application scenario, namely VidTAB for few-shot classification and VidEB for zero-shot retrieval. The paper evaluated a number of publicly available image and video foundation models and provides some observations based on the benchmark results.

Pros:
* The paper propose to extend the evaluation datasets and evaluation methods based on existing works.
* The paper adds more publicly available foundation models to benchmark.

Cons:
* Some selection of datasets and evaluation protocols are not clearly justified.
* Some claims made by the paper maybe not be well upheld by the experiments.

Please see the below session for details.

**Strengths:**

* This paper broadens the scope of existing video benchmark suites by integrating a wider range of tasks, contributing to the development of a more comprehensive evaluation framework.
* The authors' comprehensive evaluation of numerous publicly available vision foundation models is commendable. The resulting data serves as a valuable reference and facilitates analysis of the current research landscape.
* This work enhances existing video benchmark suites by incorporating evaluation cost as a key factor, offering significant practical advantages.

**Weaknesses:**

* Justification of dataset selection 1: In VidTAB, why "Action Recognition in Special Scenarios" are focused instead of accessing general action recognition? Furthermore, it is not clear to me why "dark scene" is representative in action recognition in special scenarios. More justifications like reference to related works are needed.
* Justification of dataset selection 2: I have concerns regarding the validity and rationale behind the dataset choices in VidEB, particularly the focus on scene retrieval tasks. The superior performance of image-based foundation models on these benchmarks suggests a potential bias towards static scene understanding, which may not fully capture the desired motion and dynamic understanding capabilities expected in a video context.
* Following data selection 2: as the evaluation sets are towards static scene understanding, I found it hard to ground the observations and conclusion in VidEB section on video foundation models. Maybe one way to improve the task is to re-purpose some video-centric dataset for embedding retrieval task, e.g. using nearest neighbor retrieval for zero-shot classfication.
* Model sizes in the comparison in the paper: in the main table (Tab 5, 6), foundation models of different sizes are put together to compare. As it largely follows the intuition that the larger the model sizes the better the performance, I found it hard to get insights when comparing models of different size in the table.
* It is strange to see the "zero-shot performance of visual languange models" in table 5. With TA-score proposed in eq. 1, how the score on zero-shot V-L model performance is calculated? as I understand V-L retrieval does not involve in few-shot examples.
* [minor] Table 2 characters are too small to read.

**Questions:**

* L482 "For VidTAB, to ensure fair comparison and efficient assessment, we train all models for the same number of epochs and made minor adjustments to the hyperparameters to ensure convergence." I am not convinced that minor adjustments to the hyperparameters would lead to a fair comparison to the model performance, especially on the few-shot setup. It would be valuable to provide the hyperparameters tuning details and make sure all models reach the best performance, for a fair comparison and conclusion.

---

### Note · Authors · 2024-11-14

I have read and agree with the venue's withdrawal policy on behalf of myself and my co-authors.